# Transcriptome analysis and identification of genes associated with floral transition and fruit development in rabbiteye blueberry (*Vaccinium ashei*)

**Xuan Gao**[1,☯], **Lida Wang**[1,☯], **Hong Zhang**[1,2], **Bo Zhu**[1], **Guosheng Lv**[1], **Jiaxin Xiao**[1] *

**1** Anhui Provincial Key Laboratory of Molecular Enzymology and Mechanism of Major Diseases and Key Laboratory of Biomedicine in Gene Diseases and Health of Anhui Higher Education Institutes, Anhui Normal University, Wuhu, Anhui, China, **2** Anhui Microanaly Gene Limited Liability Company, Hefei, Anhui, China

☯ These authors contributed equally to this work.
* xjx0930@163.com

**Data Availability Statement:** All relevant data are within the manuscript and its Supporting Information files.

## Abstract

Flowering and fruit set are important traits affecting fruit quality and yield in rabbiteye blueberry (*Vaccinium ashei*). Intense efforts have been made to elucidate the influence of vernalization and phytohormones on flowering, but the molecular mechanisms of flowering and fruit set remain unclear. To unravel these mechanisms, we performed transcriptome analysis to explore blueberry transcripts from flowering to early fruit stage. We divided flowering and fruit set into flower bud (S2), initial flower (S3), bloom flower (S4), pad fruit (S5), and cup fruit (S6) based on phenotype and identified 1,344, 69, 658, and 189 unique differentially expressed genes (DEGs) in comparisons of S3/S2, S4/S3, S5/S4, and S6/S5, respectively. There were obviously more DEGs in S3/S2 and S5/S4 than in S4/S3, and S6/S5, suggesting that S3/S2 and S5/S4 represent major transitions from buds to fruit in blueberry. GO and KEGG enrichment analysis indicated these DEGs were mostly enriched in phytohormone biosynthesis and signaling, transporter proteins, photosynthesis, anthocyanins biosynthesis, disease resistance protein and transcription factor categories, in addition, transcript levels of phytohormones and transporters changed greatly throughout the flowering and fruit set process. Gibberellic acid and jasmonic acid mainly acted on the early stage of flowering development like expression of the florigen gene *FT*, while the expression of auxin response factor genes increased almost throughout the process from bud to fruit development. Transporter proteins were mainly associated with minerals during the early flowering development stage and sugars during the early fruit stage. At the early fruit stage, anthocyanins started to accumulate, and the fruit was susceptible to diseases such as fungal infection. Expression of the transcription factor MYB86 was up-regulated during initial fruit development, which may promote anthocyanin accumulation. These results will aid future studies exploring the molecular mechanism underlying flowering and fruit set of rabbiteye blueberry.

**Funding:** This work was supported by the National Natural Science Foundation of China (31101148) and the foundation of Anhui Normal University (2010cxjj14). The funders had no role in study design, data collection and analysis, decision to publish, or preparation of the manuscript.

**Competing interests:** The authors have declared that no competing interests exist.

## Introduction

Flowering and fruit set are two of the most important events in the plant life cycle. The flowering process is a transition from vegetative growth to reproductive development, which directly affects fruit yield and quality. Falling flowers and fruits often lead to poor harvests. Understanding flowering and fruit set in blueberries will facilitate the improvement of fruit traits and yield.

Flower development is the most important developmental event in the life cycle of higher plants. The flowering pathway has been well studied in Arabidopsis (*Arabidopsis thaliana)* and to some extent in other monocots and dicots. Transition to flowering is controlled by complex genetic networks including the vernalization pathway [1], autonomous pathway [2], hormone pathway [3], and photoperiodic pathway [4]. In Arabidopsis, the key genes integrating multiple flowering pathways are *FLOWERING LOCUS T (FT)*, *SUPPRESSOR OF OVEREXPRESSION OF CONSTANS1 (SOC1)*, and *LEAFY (LFY)*. *FT* which is a florigen gene and *SOC1* share the common upstream regulators *CONSTANS (CO)*, a key component of the photoperiodic pathway, and *FLOWERING LOCUS C (FLC)*, a flowering repressor integrating autonomous and vernalization pathways. Rice (*Oryza sativa*), a model plant for monocotyledon molecular genetics, has two florigen genes, *Headingdate3a (Hd3a)* and *RICEFLOWERINGLOCUST 1 (RFT1)* [5]. There are two main genetic pathways in the photoperiod flowering pathway in rice: the evolutionarily conserved OsGI–Hd1(heading date 1)–Hd3a pathway (similar to the GI–CO–FT pathway in Arabidopsis) and the Ehd1(early heading date 1)-centered specific pathway, both of which are eventually integrated into two florigen genes [5]. The phytohormone gibberellic acid (GA) seriously affects the development of male reproductive organs in plants. In Arabidopsis and rice, low gibberellin content leads to stamen defects and male sterility [6, 7]. The flowering pathway of woody plants shows differences from those of Arabidopsis and rice in some respects. *VERNALIZATION INSENSITIVE (VIN) and SHORT VEGETATIVE PHASE (SVP)* transcriptionally inhibit the expression of *FT* in Arabidopsis, while they combined with *SOC1* to induce *FT* expression in *Chimonanthus praecox* [8]. Fruit trees require chilling, which can promote flowering in *Ziziphus jujuba* [9], but is not a necessary condition for flowering. Furthermore, the expression of *C-REPEAT BINDING FACTOR (CBF)* is not significantly changed during chilling in pear [10]. Phytohormones also play an import role in regulating flowering in woody plants. Of these phytohormones, GA regulates plant flowering time; however, the genes related to GA biosynthesis have different expression patterns in sweet apple [11]. Research on *Vaccinium* species (blueberry) has been conducted over the past a few years. Transgenic blueberry with overexpression of *Vaccinium corymbosum FLOWERING LOCUS T (VcFT)* has no chilling requirement for normal flowering [12], although transgenic plants are smaller than untransformed plants. This suggests that *VcFT* is a primary integrator, but hinders the growth development of blueberry. By comparing the transcriptome and metabolome of leaves, flower buds, and mature flowers of 'Mu-Legacy' blueberry, many differentially expressed genes (DEGs) associated with cytokinin and GA biosynthesis and signaling were identified [13]. Overexpression of *DWARF AND DELAYED FLOWERING 1 (VcDDF1)*, which is associated with delayed flowering in Arabidopsis, can increase freezing tolerance with no effect on plant flowering time in blueberry [14].

Previous transcriptome analysis showed that the genes involved in phenylpropanoid biosynthesis and nitrogen metabolism, as well as cutin, suberin, and wax biosynthesis are up-regulated mainly in early blueberry fruit development, while genes involved in starch and sugar metabolism are highly expressed at fruit ripening [15]. Cyanogenic glycoside (CG) biosynthetic enzymes are highly expressed in green fruit, but a CG detoxification enzyme is up-regulated during fruit ripening [16]. Genes related to sugar and organic acids are up-regulated

during fruit development [17]. Genome-wide analysis showed that high expression levels of *Vaccinium corymbosum* caffeic acid O-methyltransferase (VcCOMT) family genes during the development of blueberry fruit may contribute to greater fruit firmness [18]. The process of transition from flower to fruit involves the anthocyanin pathway, transcription factor-related genes, sugar-related genes, hormone-related genes, and photosynthesis. Flavonoids accumulate from the early fruit stage to the maturation stage [19]. High levels of expression of genes related to anthocyanin and carotene biosynthesis [20] may lead to pigment changes during fruit development. Some transcription factor-related genes such as MYB transcription factor (*MYB*) also have high expression levels at the fruit development stage. Glucose accumulates promptly during fruit development [19, 20]; however, the expression of *SUCROSE TRANS-PORTER 2 (SUC2)* and *SWEET1*, which encode sugar transporters, is down-regulated [19]. Hormone pathways display complex regulation at this stage. Genes related to ethylene [19] and cytokinins [11, 20] are up-regulated when plants transition from flower to fruit. On the contrary, genes related to IAA and auxin are down-regulated [11]. The regulation of GA is particularly complicated during the flowering process. Expression of *GA2OX* is up-regulated [19], while expression of some other genes is down-regulated [11]. Photosynthesis may not matter during the transformation stage, as evidenced by many genes related to photosynthesis being expressed at lower levels from flowering to fruit set [11].

Many researchers have reported on changes in gene expression at the flower stage and fruit stage; however, few researchers have reported on the relationship between flowering and fruit set in blueberry. Gene expression patterns may be unique at different stages from bud to fruit. For example, *SVP* and *MAFs*, which are involved in vernalization, have a high expression level from the bud to the flower stage [21] but this decreases later. Blueberry yield depends not only on pollen performance but also on genes related to hormones and on nutrition supplementation during the flower-fruit transition. In the present study, we generated RNA-seq data for five different stages of blueberry flower development from bud to fruit. We took advantage of techniques employed in Cufflinks using the genomic sequence of blueberry published in 2019 [15]. We aimed to determine the critical genes at each stage of blueberry development from bud to fruit, which will provide new insight into the mechanism of flowering and fruit set in blueberry and enable effective breeding of flowering and fruit set in blueberries.

## Results

### Identification and functional annotation of DEGs

We explored the expression levels of genes involved in blueberry flower and fruit set at six stages (Fig 1A) using Cuffdiff software instead of edgeR. After filtering (false discovery rate FDR <0.05) (S1 Table), a total of 34 genes were differentially expressed, among which seven were down-regulated and 27 were up-regulated, in the S2 stage compared with the S1 stage (leaf bud to flower bud) (S2 Table). A total of 2,298 genes showed differential expression, with 1,389 up-regulated and 909 down-regulated, in S3 compared with S2 (flower bud to initial flower) (Fig 1C and S3 Table). The comparison of S4 with S3 (initial flower to bloom flower) revealed 453 DEGs, with 180 up-regulated and 273 down-regulated (Fig 1C and S4 Table). A total of 1,692 genes displayed differential expression in S5 compared with S4 (bloom flower to pad fruit), with 784 up-regulated and 908 down-regulated (Fig 1C and S5 Table). The comparison of S6 with S5 (pad fruit to cup fruit) identified 621 genes showing differential expression, with 323 up-regulated and 298 down-regulated (Fig 1C and S6 Table). The number of DEGs in S3/S2 and S5/S4 was dramatically higher than that for the other three pairs.

The Venn diagram displays a comparative analysis of the DEGs described above (Fig 1B). We identified 5,098 DEGs associated with transitions to different developmental stages from

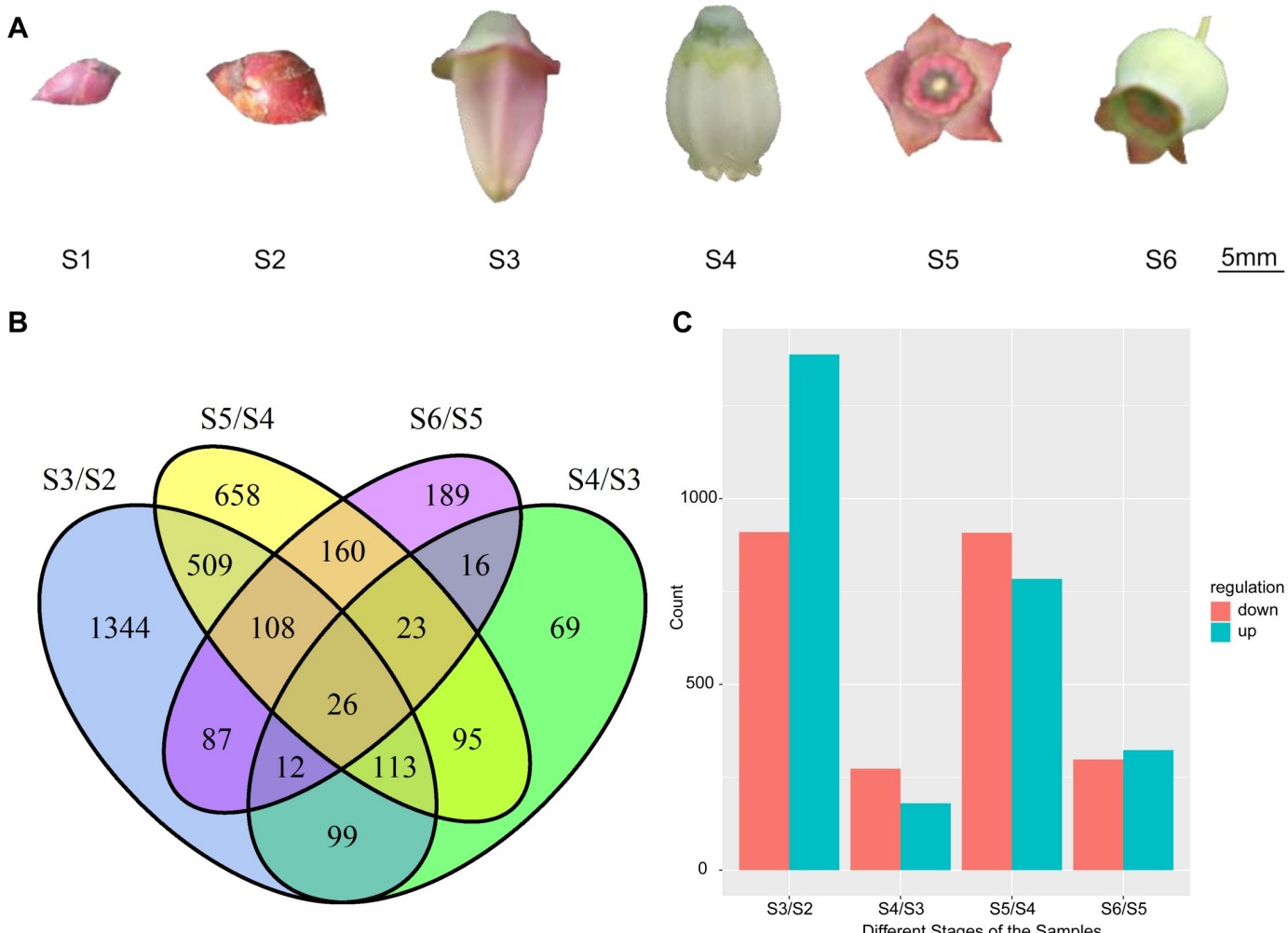

**Fig 1. Analysis of Differentially Expressed Genes (DEGs) between each library pair.** (A) Samples of *Vaccinium ashei* for RNA-seq. S1, leaf bud; S2, flower bud; S3, initial flower; S4, bloom flower; S5, pad fruit; S6, cup fruit. (B) Venn diagram of the DEGs in comparisons of five samples (numbers in circles represent the numbers of DEGs in different stages). (C) Numbers of up-regulated (up) and down-regulated (down) genes in the comparisons of five samples.

bud to fruit setting, with 1,344, 69, 658, and 189 DEGs being unique to S3/S2, S4/S3, S5/S4, and S6/S5, respectively (Fig 1B and S7–S10 Tables). These data showed that there were obviously more DEGs at S3/S2 and S5/S4 than at S4/S3, and S6/S5, suggesting that S3/S2 and S5/S4 represented major transitions from buds to fruit in blueberry.

## Principal component analysis

Our principal component analysis (PCA) shown in Fig 2 indicated that Comp.1 (30.37%) and Comp.2 (16.95%) best described the sources of variance between the samples at different stages. In general, replicate samples belonging to a given developmental stage clustered more closely together than samples belonging to different stages, except for leaf bud and flower bud (Fig 2). This result is consistent with the number of DEGs identified. As shown above, 2,298 DEGs were identified between S2 and S3 stages, which was dramatically higher than the number between other stages (Fig 1C and S3 Table).

PCA analysis of all the transcripts in 18 samples revealed that those belonging to stages S2, S3, S4, S5, and S6 were well separated from each other, indicating that gene expression patterns

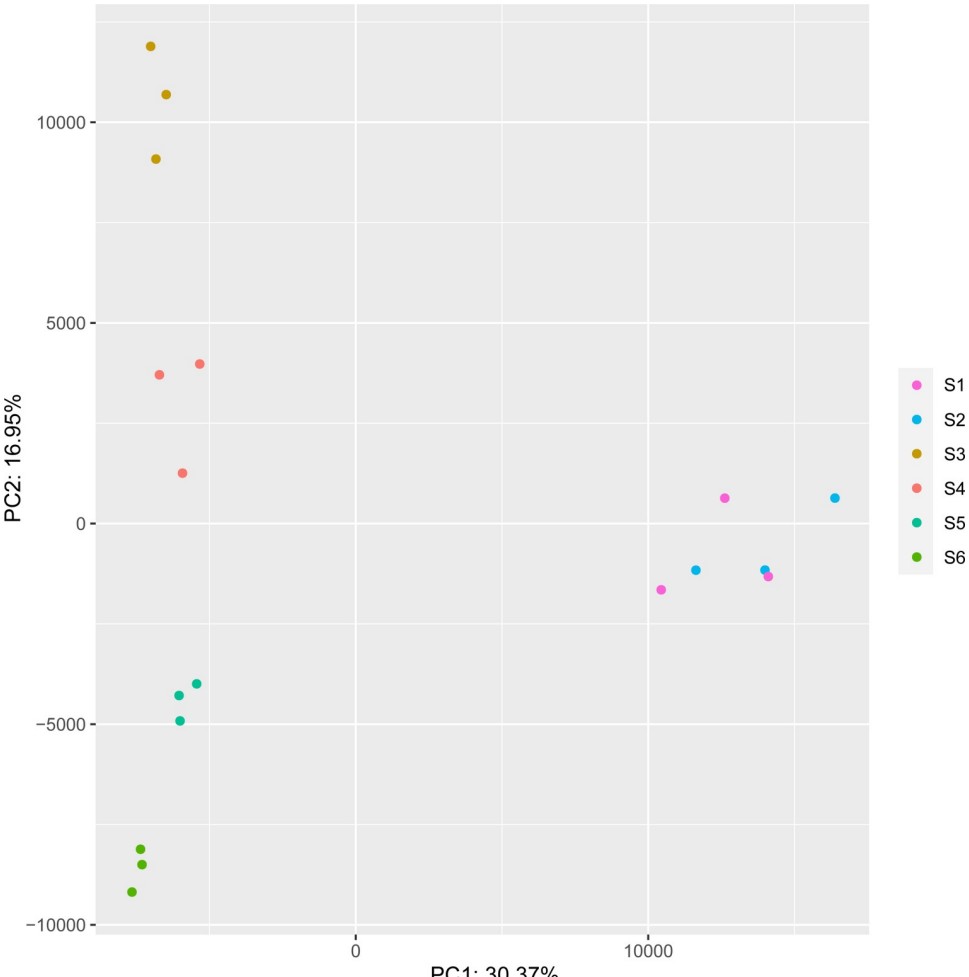

**Fig 2. Principle Component Analysis (PCA) of variability in DEGs observed across samples.** Colors are used to differentiate between samples: purple, blue, brown, red, green, and cyan represent leaf bud, flower bud, initial flower, bloom flower, pad fruit, and cup fruit, respectively.

were different between buds, flowers, and fruit. However, S1 and S2 were not separated from each other very well, because leaf bud and flower bud had similar phenotypes, which may lead to accidental mis-sampling or misidentification. The process from leaf buds to flower buds does not represent two consecutive developmental stages as well, for that leaf buds will differentiate into either leaf or flower buds. What's more, the number of DEGs between S1 and S2 was only 34 genes (S2 Table). To make the results more accurate, we have removed S1 sample in our analysis above and below.

## GO analysis for different flower developmental stages

To determine the biological classification of DEGs, we performed functional and pathway enrichment analyses using ClusterProfiler R packages. Gene ontology (GO) analysis was performed by stage. Comparing S3 with S2, 2,298 (S3 Table) DEGs were significantly enriched in "aromatic compound catabolic process", "cytoskeleton organization", and "vegetative to reproductive phase transition of meristem" in the Biological Process category (Fig 3A). The GO terms "vesicle" and "thylakoid" were prominent in the Cellular Component category, whereas

"adenyl nucleotide binding" and "active transmembrane transporter activity" were the apparent molecular functions of the DEGs. Comparing S4 with S3, "response to jasmonic acid", "programmed cell death", and "response to organonitrogen compounds" were notable under the Biology Process category (Fig 3B). In the comparison of S5 with S4, "mitotic cell cycle", "cytoskeleton organization", and "cell division" were dominant terms in the Biology Process category (Fig 3C). All these GO terms were related to cell growth, inferring that the number of cells began to rise at this stage. Abiotic and biotic stress related pathways also played an important role at the S5/S4 stage. Abiotic stress includes "response to water", "response to light intensity", and "response to heat" while biotic stress includes "response to virus". The "flavonoid biosynthetic process" was another dominant term in pad fruit compared with bloom flower, indicating that flavonoids began to accumulate from the bloom stage to the early fruit stage. In the Cellular Component category, "intrinsic component of membrane" and "plant-type cell wall" were predominant terms. For the Molecular Function category, "ATP binding" and "adenyl nucleotide binding" were the most strikingly enriched terms. In the comparison of S6 with S5, "mitotic cell cycle", "cytoskeleton organization", and "cell division" were dominant terms in the Biology Process category, mostly similar to the bloom flower/pad fruit transition (Fig 3D). The difference between these comparisons was that stages S5 and S6 focused on organ development such as "regulation of post-embryonic development" and "regulation of shoot system development".

GO analysis indicated that DEGs enriched at the early flowering development stage (S3/S2) were involved in similar functions to those enriched at the early fruit development stage (S5/S4 and S6/S5), such as "cytoskeleton organization". The difference was that the S3/S2 stage DEGs were also enriched in "aromatic compound catabolic process" and "vegetative to reproductive phase transition of meristem", preparing for flowering, while the S5/S4 and S6/S5 DEGs were mainly concentrated in "abiotic and biotic stress"-related pathways and "flavonoid biosynthesis", which was consistent with the results of the following kyoto encyclopedia of genes and genomes (KEGG) analysis.

## DEGs related to flowering process in blueberry

We further conducted an overview hierarchical clustering analysis of the DEGs at each stage. DEGs in different replicates of the same stage exhibited similar expression patterns. However, comparisons of the same DEGs at different stages revealed differences in expression patterns (Fig 4).

DEGs related to transporters, phytohormone biosynthesis and signaling, and photosynthesis showed considerable changes in expression from bud stage to initial flower stage (S3/S2) (Fig 4A). Some transfer genes were up-regulated at this developmental stage, such as those encoding the MATE efflux family protein (*MATE*), heavy metal transport/detoxification superfamily protein and high affinity K+ transporter 5 (*HAK5*), indicating that blueberry requires mineral elements during the transition from buds to initial flowers. Expression of genes encoding sugar transporter proteins (*SUT4* and *STP7*) was also up-regulated, while that of genes encoding nitrate transporters (*NRT1.5*) was down-regulated, suggesting that an increase in C/N ratio is beneficial for flower bud formation and flowering. This was consistent with findings in orchid, where C/N ratio played an important role in shaping structure of flowers [22]. DEGs related to GA (*GASA1*), auxin (*AUX2*, SAUR–like auxin–responsive protein family), jasmonic acid (*JAZ1*), and abscisic acid (ABA) (*ABA3*, *ABA4*) were up-regulated while genes related to ethylene (*ERF1*) and IAA (*IAA11*, *IAA29*) were down-regulated. DEGs related to photosynthesis such as *AGAMOUS-like 20* (*AGL20*) and *chlorophyll A−B binding family protein* (*CAB*) were down-regulated.

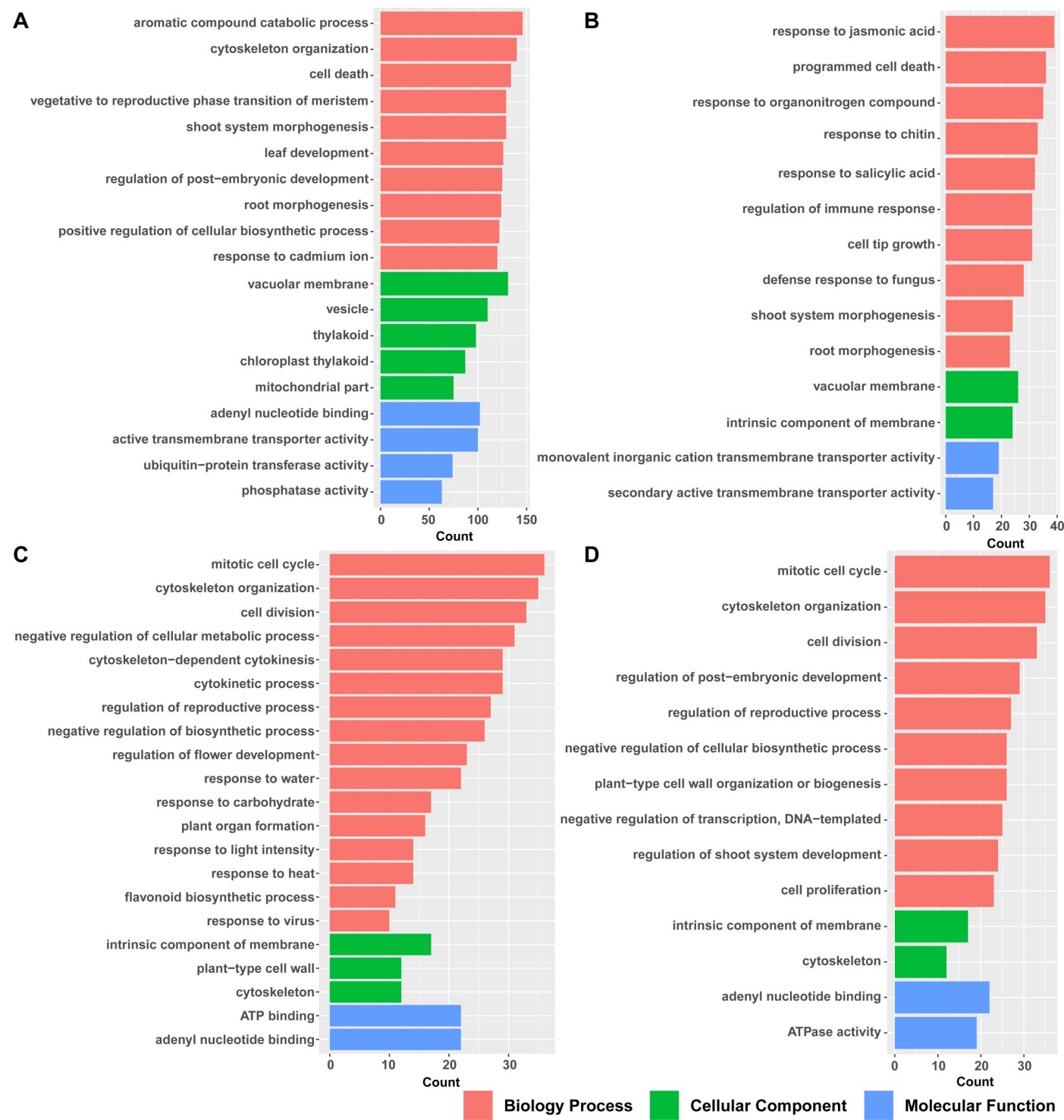

**Fig 3. GO analysis between developmental stages in blueberry.** (A) S3/S2; (B) S4/S3; (C) S5/S4; (D) S6/S5. S2, S3, S4, S5, and S6 represent flower bud, initial flower, bloom flower, pad fruit, and cup fruit, respectively.

A great many DEGs related to phytohormone signaling such as auxin, GA, and JA showed expression changes from initial flowers to bloom flowers (S4/S3) (Fig 4B). More interestingly, all phytohormones related genes were down-regulated in this stage except *auxin response*

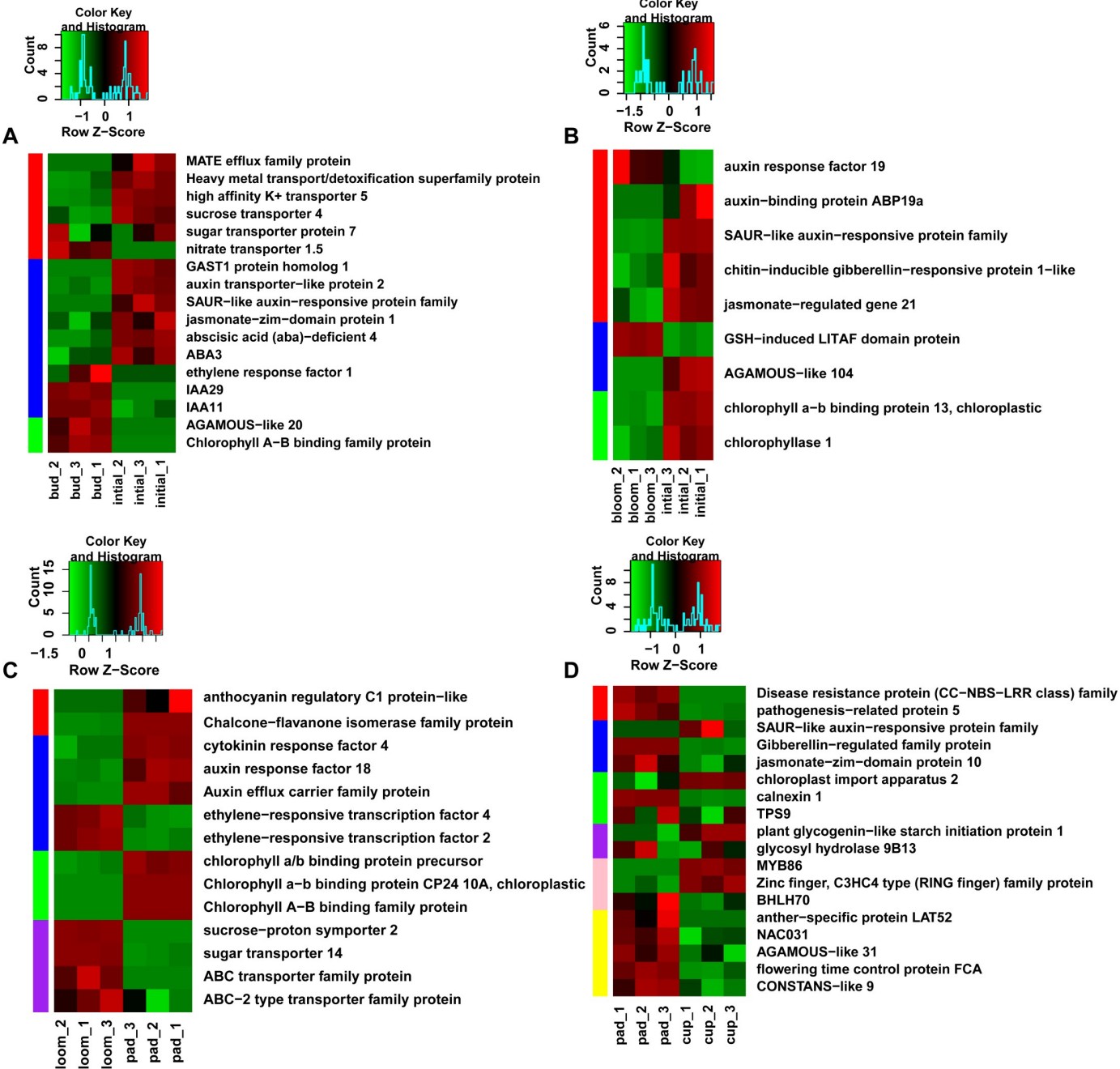

**Fig 4. Gene expression profiles of differentially expressed genes in four comparisons.** (A) S3/S2; (B) S4/S3; (C) S5/S4; (D) S6/S5. S2, S3, S4, S5, and S6 represent flower bud, initial flower, bloom flower, pad fruit, and cup fruit, respectively. Green represents genes down-regulated and red represents genes up-regulated. The abbreviations and full names of the genes involved are shown below: *HAK5* (*high affinity K+ transporter 5*), *MATE* (*multidrug and toxic compound extrusion efflux family protein efflux family protein*), *SUT4* (*sucrose transporter 4*), *STP7* (*sugar transporter protein 7*), *NRT1.5* (*nitrate transporter 1.5*), *GASA1* (*GAST1 protein homolog 1*), *AUX2* (*auxin transporter–like protein 2*), *JAZ1* (*jasmonate–zim–domain protein 1*), *ABA4* (*abscisic acid (aba)–deficient 4*), *ABA3* (*molybdenum cofactor sulfurase (LOS5)*), *ERF1* (*ethylene response factor 1*), *IAA29* (*indole-3-acetic acid inducible 29*), *IAA11* (*indole-3-acetic acid inducible 11*), *AGL20* (*AGAMOUS–like 20*), *CAB* (*chlorophyll A–B binding family protein*), *ARF19* (*auxin response factor 19*), *ABP19A* (*auxin–binding protein ABP19a*), *JRG21* (*jasmonate–regulated gene 21*), *GILP* (*GSH–induced LITAF domain protein*), *AGL104* (*AGAMOUS–like 104*), *CLH1* (*chlorophyllase 1*), *CHI* (*chalcone–flavanone isomerase family protein*), *CRF4* (*cytokinin response factor 4*), *ARF18* (*auxin response factor 18*), *ERF4* (*ethylene–responsive transcription factor 4*), *ERF2* (*ethylene–responsive transcription factor 2*), *CAB-10A* (*chlorophyll a–b binding protein CP24 10A*, chloroplastic), *SUC2* (*sucrose–proton symporter 2*), *STP14* (*sugar transporter 14*), *ABC* (*ABC transporter family protein*), *ABCG4* (*ABC–2 type transporter family protein*), *PR5* (*pathogenesis–related protein 5*), *JAZ10* (*jasmonate-zim-domain protein 10*), *CIA2* (*chloroplast import apparatus 2*), *CNX1* (*calnexin 1*), *TPS9* (*trehalose-phosphatase/synthase 9*), *PGSIP1* (*plant glycogenin–like starch initiation protein 1*), *GH9B13* (*glycosyl hydrolase 9B13*), *MYB86* (*myb domain protein 86*), *BHLH70* (*basic helix-loop-helix 70*), *LAT52* (*anther–specific protein LAT52*), *NAC031* (*Protein CUP-SHAPED COTYLEDON 3*), *AGL31* (*AGAMOUS–like 31*), *FCA* (*flowering time control protein FCA*), *COL9* (*CONSTANS–like 9*).

*factor 19* (*ARF19*). The DEGs related to pollen growth such as *AGAMOUS-like 104* (*AGL104*) were expressed lower. The DEGs related to photosynthesis such as *chlorophyll a-b binding protein 13* and *chlorophyllase 1* (*CLH1*) were down-regulated as well.

DEGs related to anthocyanin biosynthesis, phytohormone signaling, photosynthesis, and sugar transporters showed significant changes in expression from bloom flowers to pad fruit (S5/S4) (Fig 4C). Genes encoding anthocyanin regulatory C1 protein−like and CHI (Chalcone −flavanone isomerase family protein) were expressed more highly in pad fruit than at the bloom stage. Some genes related to cytokinins (*CRF4*) and auxin (*ARF18* and *auxin efflux carrier family p*rotein) were up-regulated, and some genes related to ethylene (*ethylene-responsive transcription factor 2* and *ethylene-responsive transcription factor 4*) were down-regulated (Fig 4C). Down-regulation of genes encoding ethylene-responsive transcription factors at this stage may reduce the content of ethylene to avoid early fruit falling. DEGs related to photosynthesis (*chlorophyll a/b binding protein precursor*, *CAB-10A*, and *CAB*) were expressed at higher levels in pad fruit than in bloom flowers. Genes encoding transporters such as sugar transporters (*SUC2* and *STP14*) and ABC transporters (*ABC* and *ABCG4*) were all down-regulated in pad fruit compared with bloom flowers. What was particularly interesting was that expression of phytohormone genes related to auxin was increased during both the bud to flower transition and the flower to fruit transition (Fig 4A and 4C), while expression of genes encoding transporters for sugar was increased at the bud to flower transition (*SUT4* and *STP7*) (Fig 4A) but decreased at the flower to fruit transition (*SUC2* and *STP14*) (Fig 4C).

Expression levels of DEGs related to pathogenesis, phytohormone signaling, photosynthesis, sugars, transcriptional factors, and floral organ development were changed significantly in cup fruit (S6) compared with pad fruit (S5) (Fig 4D). Genes encoding disease-resistance proteins and pathogenesis-related protein (*PR5*) were down-regulated in cup fruit compared with pad fruit. DEGs related to GA (*Gibberellin−regulated family protein*) and JA (*JAZ10*) were also down-regulated, while auxin-related genes such as SAUR-like auxin-responsive protein family were mainly up-regulated throughout the whole process from bud to fruit. Transporter genes such as those encoding MYB86 and Zinc finger/C3HC4-type family protein were up-regulated while that encoding BHLH70 was down-regulated at the cup fruit stage compared with the pad fruit stage. *NAC031* is involved in the molecular mechanisms regulating shoot apical meristem (SAM) formation and acts as an inhibitor of cell division [23]. *AGAMOUS-like 31* (*AGL31*) acts as a floral repressor [24]. *Flowering time control protein FCA* regulates flowering time, seed size, and cell volume. *CONSTANS-like 9* (*COL9*) may be involved in the light input to the circadian clock [25]. All of these genes, *NAC031*, *AGL 31*, *FCA*, and *COL9*, were down-regulated during the pad fruit to cup fruit transition, indicating that the flowering process was just completed at this stage.

Changes in DEGs related to phytohormone biosynthesis and signaling, transporter proteins, photosynthesis, anthocyanin biosynthesis, disease resistance proteins, and some transcription factors (TFs) represented the major changes during the development from bud to fruit.

## Validation of DEGs by quantitative reverse-transcription PCR (qRT-PCR)

To confirm the accuracy of the RNA-seq results, we examined the transcription of nine putative genes involved in flower development through a supporting qRT-PCR experiment (Fig 5). Transcriptional levels of *AUX2* (*auxin transporter-like protein 2*) and *CHI* (*chalcone-flavanone isomerase family protein*) were significantly up-regulated (p<0.05) during the process of flowering and fruit set compared with earlier developmental stages, in agreement with the alterations in gene expression detected by the transcriptome analysis. The transcriptional level of

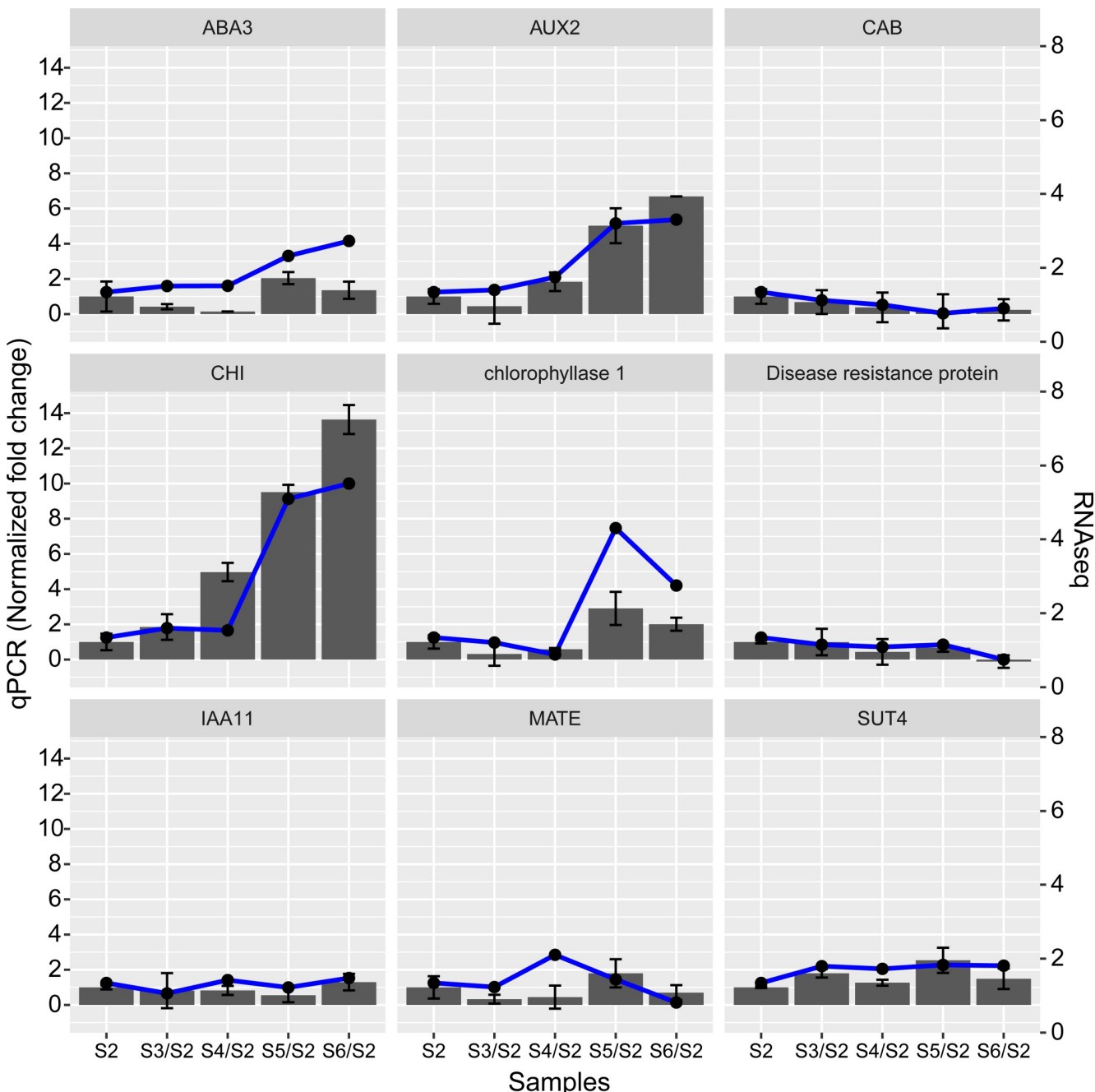

**Fig 5. Validation of gene expression by qRT-PCR.** Expression levels of nine candidate genes—*ABA3* (*molybdenum cofactor sulfurase* (*LOS5*)), *AUX2* (*auxin transporter-like protein 2*), *CAB* (*chlorophyll A-B binding family protein*), *CHI* (*chalcone-flavanone isomerase family protein*), *CLH1* (*chlorophyllase 1*), *Disease resistance protein* (CC-NBS-LRR class) family member, *IAA11* (*indole-3-acetic acid inducible 11*), *MATE* (encoding a MATE efflux family protein), and *SUT4* (*sucrose transporter 4*)—were determined at each stage using qRT-PCR analyses. S2, S3/S2, S4/S2, S5/S2 and S6/S2 represent flower bud, initial flower, bloom flower, pad fruit and cup fruit, respectively. The bars and error bars (Standard Deviation) represent the qPCR while the line represents the results of RNA-seq.

*CAB* (encoding chlorophyll A-B binding family protein) decreased from the bud stage to pad fruit and then increased slightly again in cup fruit. The transcriptional level of *chlorophyllase I* peaked at the pad fruit stage and then decreased. These nine genes provided more evidence to support the reliability of our RNA sequencing data.

## Discussion

Uncovering the rules for flowering and fruit set in blueberry will facilitate improvement of fruit traits. The rapid advances and reduction in cost of sequencing offer exclusive opportunities for exploring the molecular mechanisms underlying flowering and fruit set processes in blueberry species. Furthermore, recent publication of the genomic sequence of blueberry increases the accuracy of transcriptome analysis [15]. Previous studies have only focused on three stages during the development of flowering in blueberry [26]. The lack of comparative transcriptome analysis between different stages of flowering and fruit set in blueberry impedes further functional genomics and molecular biology research. In this study, we divided flowering and fruit set into five phases distinguished by the morphology of flowers, including bud stages, flower stages, and fruit stages.

Comparing the number of DEGs at different stages, we found significantly more genes regulated from flower bud to flower and from flower to fruit than during other stage transitions (Fig 1B and 1C). The genes differentially expressed during these two transitions may play important roles in reproductive growth in blueberry.

Aromatic compound catabolic process pathways are involved in the breakdown of aromatic secondary metabolites, which have high content in plants. DEGs were remarkably enriched in the "aromatic compound catabolic process" from bud to initial flower stages indicating that a large number of secondary metabolites began to accumulate (Fig 3A). ABA can enhance the accumulation of *betaine aldehyde dehydrogenase 2* (*BADH2*), which is related to flower fragrance in most plants [27]. In this study, *ABA3*, which acts as a key regulator of ABA biosynthesis [28], was up-regulated in initial flowers of blueberry (Fig 4A). We speculated that aromatic compounds were increased through regulating the content of ABA. Phytohormone-related genes have complicated roles throughout the flowering and fruit set period. Phytohormones contribute to flower development, such as outgrowth of organs, development of male and female gametophytes and cell elongation. The gene encoding Gibberellin-regulated protein 1 (*GAST1*) [29], which may be involved in seed germination, flowering, and seed maturation was up-regulated in our study. IAA11 proteins act as repressors of early auxin response genes [30]. Down-regulation of *IAA11* and up-regulation of genes encoding auxin transporter proteins suggests that the auxin content increased during this stage. Auxin response genes were up-regulated, consistent with previous reports [31], showing the importance of auxin at the initial flower stage. Photosynthesis-related genes such as *chlorophyll A-B binding family* protein (*CAB*) were down-regulated, consistent with previous reports [11], indicating that photosynthesis may have little effect on the transition from buds to initial flowering. Consistent overexpression of FT protein may impede further development of blueberry flowers [32]. *AGAMOUS-like 20* (*AGL20*), involved in the FT-regulated pathway, may integrate signals from the photoperiod, vernalization, and autonomous floral induction pathways [33]. In our study, we identified no significant difference in expression of genes related to *FT*, except *AGL20*, from the bud to initial flower stages (Fig 4A). Expression of *AGL20* decreased, suggesting that the flowering process was regulated by *FT* just before the initial flower stage or even earlier. Overexpression of *FT* may lead to abnormal flowering in blueberry [12]. Photoperiod and florigen *FT* have little effect on flower development at this stage, and *FT* can even prevent flowering. This was similar to GA and jasmonic acid (JA) in our study (Fig 4). GA can promote vegetative growth to reproductive growth, but it can inhibit growth from the initial stage of reproductive growth to the flowering stage [34]. JA produces herbivore-induced volatiles to attract predatory insects [35], in preparation for attracting insects.

The transition from flowering to fruit set is another important transition in blueberry. Genes encoding Anthocyanin regulatory C1 protein-like and CHI, both involved in the

anthocyanin biosynthesis pathway [36, 37], were upregulated during this transition. Anthocyanins may well set out to synthesize from pad fruit stage for the content of anthocyanins can be detected in green fruit in our previous and others' research [38, 39]. We discovered that photosynthesis increases from initial flower stage to pad fruit stage, suggesting that supplementation of carbohydrate accumulation is required for growth of young fruit. Despite plentiful evidence for functional photosynthesis in young fruit [40, 41], studies on tomato have shown that fruit photosynthesis was not necessary for fruit development but beneficial for seed development, at least under ambient conditions [41].

From the pad fruit to the cup fruit stage, the fruit expands quickly and then remains in the cup stage for a long time. There were many similar results between our research and that of Colle et al. [15] at this stage. In our research, "mitotic cell cycle", "cytoskeleton organization", and "cell division" were the main GO terms associated with DEGs between pad fruit and cup fruit (Fig 3D), indicating that genes associated with cell division and cell wall biosynthesis were differentially expressed during the earliest fruit developmental stage. This result was consistent with Colle's reports [15]. Genes regulating defensive response-related genes were also highly up-regulated during young fruit development in the research of Colle [15]; however, in our study, expression of genes encoding the disease resistance protein (CC-NBS-LRR class) family and pathogenesis-related protein 5 decreased significantly form pad to cup fruit (Fig 4D). Rabbiteye blueberry are susceptible to infection by the fungus *Exobasidium maculosum* on young fruit [42], while mature fruit tends to be infected by blueberry red ringspot virus (BRRV) during the ripening period in the spring [43]. Furthermore, bluberry fruits infected by *Monilinia vaccinia-corymbosi* are unfit for processing [44]. We speculated that rabbiteye blueberry may well be vulnerable to fungal infection at the cup fruit stage. The expression of genes encoding disease resistance protein increased from bloom to pad fruit according to our qRT-PCR results (Fig 5). The difference between the results obtained by Colle et al. and our results may be due to the sampling time.

Most of phytohormone related gene were down-regulated from initial flower stage to bloom stage. The *AGL104* involves in the regulation of pollen maturation and pollen tube growth [45]. The expression of *AGL104* decreased (Fig 4B) indicated that pollen has already been developed during the initial flower stage. And *MYB86* gene expression was up-regulated during initial fruit development (Fig 4D). *MYB86* expression was found to be concurrent with anthocyanin accumulation in fruits of *Fragaria vesca* [46]. Therefore, *MYB86* may function as anthocyanin accumulation in blueberry. However, this needs further research to confirm.

In this study, we demonstrated that phytohormones, anthocyanin, and pathogenesis are involved in the blueberry flowering and fruit set process. Phytohormones may have a complex function because expression of genes encoding hormones changed frequently at different developmental stages. For example, Expression of GA-related genes peaked at the early stage of flower development, but decreased later. Expression of auxin-related genes increased nearly throughout the process from bud to fruit, like that expression of *AUX2* increased steady through the whole flower and fruit set process in blueberry. This was identified through both RNA-seq data and qRT-PCR, suggesting that *AUX2* plays an important role in the flowering process. Expression of genes related to anthocyanin was increased during later stages of blueberry flowering while that of pathogenesis-related genes was decreased, which may suggest that anthocyanin makes up some deficiency caused by lack of pathogen resistance. However, this needs further research.

## Materials and methods

### Blueberry materials

Nine-year-old shrubs of rabbiteye 'Brightwell' blueberry (*Vaccinium ashei*) were cultivated in Nanling County, Anhui Province, China. Flowering and fruit set were divided into six stages

based on morphological characterization (Fig 1A). Leaf bud (S1) and flower bud (S2) were collected in early and late March, respectively. Initial flower (S3) and bloom flower (S4) were collected in April when the blueberry was flowering. Pad fruit (S5) was collected in late April when fruit began to form and looked similar to a pad. Cup fruit (S6) was collected in early May and represented a stage present for a long time. Considering that blueberry flowers are racemes, the developmental levels of their flowers on the top and bottom are not uniform. The samples were based on individual florets rather than whole inflorescences, ensuring that the developmental levels of samples collected at each stage were consistent. The S4 flowers were almost pollinated, and all parts of the S4 flowers in Fig 1A were sequenced, including the inferior ovary and sepals. All samples were manually picked to prevent any mechanical damage and then frozen immediately in liquid nitrogen, followed by storage at -80˚C until use. Each sample had three replicates. About 200 mg of each replication was taken, of which 100 mg was used for RNA extraction.

## Construction of cDNA library and Illumina sequencing

Total RNA was extracted using Trizol reagent (Invitrogen), and genomic DNA contamination was removed using DNase I (TaKaRa, Dalian, China). RNA quality was verified using agarose gel electrophoresis and a Bioanalyzer 2100 (Aglient Technologies, Palo Alto, CA, USA). The RIN of the 18 RNA samples was in the range from 9.0 to 9.6. For cDNA library construction, mRNA was enriched using oligo (dT) magnetic beads and then broken into smaller pieces using fragmentation buffer (GeneChip® WT Terminal labeling kit; Affymetrix; P/N 900671). cDNA libraries were sequenced using an Illumina HiSeq2000 platform at WeiFen GENE Co., ChaoHu, China to generate 150-bp paired-end reads. The raw reads data are available at the National Center for Biotechnology Information NCBI sequence read archive database (PRJNA745351). The reference genome and gene model annotation files were downloaded directly from the blueberry genome website (https://www.vaccinium.org/crop/blueberry) [15]. An index of the reference genome was built using Bowtie software (v2.2.8). Raw reads were filtered to obtain high-quality reads by removing low-quality reads with Q30 no less than 89.65% (both Q20 and Q30 were used in S1 Table). The high-quality reads were mapped onto the highbush blueberry (*Vaccinium corymbosum*) reference genome using TopHat [47]. Assembled transcripts were annotated using the Cuffcompare program from the Cufflinks package. Gene expression was quantified in terms of reads per kilobase of exon model per million mapped reads (RPKM) values. Cuffdiff (v2.1.1) was used to calculate RPKM of mRNAs. The R package cummeRbund was used to count the number of reads mapped to each gene. The RPKM value for each gene was then calculated based on the length of the gene and the number of reads mapped to it. Cuffdiff command, a part of cufflink software, was used to select the DEGs; therefore, transcripts with adjusted p-value <0.05 (significant parameter was set by default) were considered to be significantly differentially expressed.

## GO analysis and identification of DEGs

A GO functional analysis was performed to identify the biological processes most strongly represented by the DEGs. GO enrichment analysis of all the DEGs of each stage was performed using the ClusterProlifer package in R software [48]. The DEGs of each stage were generated by Cuffdiff by using database Org.At.tair.db. After identifying the DEGs between S2/S1, S3/S2, S4/S3, S5/S4, and S6/S5, respectively, a Venn diagram was constructed to show the overlap between DEGs of four comparisons (S3/S2, S4/S3, S5/S4, and S6/S5) to identify unique DEGs between each stage except S2/S1 for further analysis.

Variation in DEGs between samples was studied using PCA which was based on Fragment per Kilobase Million (FPKM). Samples were grouped using transcriptional data analyzed by Cufflinks, including all the isoforms (252,060) present in each sample, for PCA analysis. The PCA was performed using the R program.

## Validation of RNA-seq data by quantitative reverse-transcription PCR

The reliability of DEGs/transcripts identified through RNA sequencing was evaluated through quantitative RT-PCR analysis of nine selected transcripts. There were two criteria used to select the nine transcripts for qRT-PCR: (1) The transcripts were related to the phytohormone or photosynthesis pathways or encoded transporters or disease resistance protein, which played a crucial role or changed considerably during flowering and fruit set.; (2) The transcripts showed relatively high expression levels and kept stable in RNA-seq data of samples to make PCR amplification easier. Every sample used flower bud as the internal control for qRT-PCR. Actin was used as an internal control to normalize expression levels of the analyzed genes.

Reverse transcription of RNA to cDNA was performed using SuperScript II reverse transcriptase (Invitrogen). The resulting cDNA from 1 µg of RNA was diluted in water, and 1 µl of sample (25 ng) was used for each PCR. Three replicated RNA samples from the flower tissues of each blueberry stage were used. Primers were designed using Primer premier v5.0 software (Applied Biosystems, Foster City, CA, USA) (S11 Table). Quantitative RT-PCR was performed in triplicate using the SYBR Green System (Thermo Fisher). Each 25 µl reaction mixture contained 25 ng cDNA, 100 nM primers, and 12.5 µl of 2×SYBR Green master mix. Reaction conditions for all primer pairs were 95˚C for 10 min, 40 cycles of 30 s at 95˚C, 60 s at 60˚C, and 60 s at 72˚C, followed by 1 cycle of 60 s at 95˚C, 30 s at 55˚C, and 30 s at 95˚C. The specificity of the reaction for each primer pair was determined using a melting curve. Relative expression was normalized using the eukaryotic translation initiation factor 3 subunit H and was calculated using the 2−ΔΔCt method.

## Conclusions

The phytohormones play an essential role in regulating flowering and early fruit development during the blueberry flowering and fruit set process. Expression of GA-related genes was up-regulated at the initial flower stage but decreased at later stages, suggesting that regulation of GA may only be effective during the early stage of the flowering process. Expression of auxin-related genes increased almost throughout the process from bud to fruit, suggesting that auxin is vital in the flowering process. Of particular interest, expression of *FT*, an inducer of flowering, showed no significant change during the bud to fruit stage. *AGL20*, which is regulated by *FT*, was down-regulated from the bud to initial flower stage, suggesting that *FT* may only affect the flowering process at the early stage, similar to GA and JA. Anthocyanin accumulation may begin at the pad fruit stage because anthocyanin biosynthesis was up-regulated at this stage. Flowering-development-related genes were constitutively expressed after the flower bloom, inferring that the flowering process is not finished until the growth of the pad fruit. Genes encoding pathogenesis-related proteins were down-regulated at the cup fruit stage, suggesting that this blueberry may be susceptible to disease.

## Supporting information

**S1 Table. Summary of the transcriptome characteristics of blueberry flowers and fruit.** (XLSX)

**S2 Table. Differentially expressed genes between stages S2 and S1.**
(XLSX)

**S3 Table. Differentially expressed genes between stages S3 and S2.**
(XLSX)

**S4 Table. Differentially expressed genes between stages S4 and S3.**
(XLSX)

**S5 Table. Differentially expressed genes between stages S5 and S4.**
(XLSX)

**S6 Table. Differentially expressed genes between stages S6 and S5.**
(XLSX)

**S7 Table. List of differentially expressed genes unique to stages S3 and S2.**
(XLSX)

**S8 Table. List of differentially expressed genes unique to stages S4 and S3.**
(XLSX)

**S9 Table. List of differentially expressed genes unique to stages S5 and S4.**
(XLSX)

**S10 Table. List of differentially expressed genes unique to stages S6 and S5.**
(XLSX)

**S11 Table. RT-qPCR genes and primers used for validation of RNA-seq data.**
(XLSX)

## Acknowledgments

We thank International Science Editing (http://www.internationalscienceediting.com) for editing this manuscript.

## Author Contributions

**Data curation:** Lida Wang.

**Funding acquisition:** Xuan Gao.

**Resources:** Hong Zhang.

**Software:** Guosheng Lv.

**Visualization:** Bo Zhu.

**Writing – original draft:** Xuan Gao.

**Writing – review & editing:** Jiaxin Xiao.

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
