## [Decision Letter · Decision Letter 0]

30 Jun 2021

PONE-D-21-18707

Transcriptome analysis of genes associated with flowering and fruit set in rabbiteye blueberry (Vaccinium ashei)

PLOS ONE

Dear Dr. gao,

Thank you for submitting your manuscript to PLOS ONE. After careful consideration, we feel that it has merit but does not fully meet PLOS ONE’s publication criteria as it currently stands. Therefore, we invite you to submit a revised version of the manuscript that addresses the points raised during the review process.

In your revision, please address all concerns raised by both reviewers, including those included in Reviewer 1's annotated file. In particular, please deposit RNA-seq data to an appropriate repository and ensure all methodology is fully and clearly explained.

We look forward to receiving your revised manuscript.

Kind regards,

Frances Sussmilch

Academic Editor

PLOS ONE

Journal Requirements:

5. Please upload a new copy of Figures 3 and 4 as the detail is not clear. Please follow the link for more information: https://blogs.plos.org/plos/2019/06/looking-good-tips-for-creating-your-plos-figures-graphics/" https://blogs.plos.org/plos/2019/06/looking-good-tips-for-creating-your-plos-figures-graphics/

6. We note that Figure 6 in your submission contain copyrighted images. All PLOS content is published under the Creative Commons Attribution License (CC BY 4.0), which means that the manuscript, images, and Supporting Information files will be freely available online, and any third party is permitted to access, download, copy, distribute, and use these materials in any way, even commercially, with proper attribution. For more information, see our copyright guidelines: http://journals.plos.org/plosone/s/licenses-and-copyright.

a. You may seek permission from the original copyright holder of Figure 6 to publish the content specifically under the CC BY 4.0 license. 

Reviewers' comments:

Reviewer's Responses to Questions

**Comments to the Author**

1. Is the manuscript technically sound, and do the data support the conclusions?

Reviewer #1: Partly

Reviewer #2: Partly

2. Has the statistical analysis been performed appropriately and rigorously? 

Reviewer #1: No

Reviewer #2: Yes

3. Have the authors made all data underlying the findings in their manuscript fully available?

Reviewer #1: No

Reviewer #2: No

4. Is the manuscript presented in an intelligible fashion and written in standard English?

Reviewer #1: No

Reviewer #2: Yes

5. Review Comments to the Author

Reviewer #1: PONE-D-21-18707 comments

The authors report transcriptome analysis in six tissues from rabbiteye blueberry ‘Brightwell’, including fully chilled leaf buds, fully chilled flower buds, early-stage flowers, full-bloom flowers, early-stage fruits (pat fruits), and “cup” fruits. Differential expressed genes (DEGs) in the paired comparisons between different tissues were identified and the potential impacts of the DEGs were discussed. The RNA-seq data presented in this manuscript are informative for a better understanding the profile of gene expressions in different tissues such as leaf buds, flower buds, flower, and fruits. Here are my major concerns (see details for 84 comments in the attached file).

Writing: Please get a professional editing service for your manuscript.

Abstract: Please rewrite it to provide accurate information of your results.

Introduction: Please remove unnecessary information and provide a concise rationale for this study/report.

1. At line 50, I would suggest rewriting this paragraph by providing the following information: 1) Provide a summary of known; flowering pathways reported for both monocots and dicots; 2) Provide information on flowering mechanism in woody (especially deciduous) plants; 3) Briefly describe the reported information on flowering of vaccinium species. There have been a lot of information on vaccinium flowering reported in the past a few years.

2. At line 51, What do you mean "the initial stage"? Is it bud-break (of dormancy release) after chilling? Please be aware that vernalization is not equal to chilling requirement. This is why vernalization is not often used for woody plant.

3. Lines 57-66, for this information, what do you want readers to know?

4. Line 67: Is this a ref on grape related to this study?

5. Line 96-97: Do you have evidence for this statement?

Results: some descriptions are not accurate. Move result descriptions from discussion to the result section.

1. Line 105: You need to summarize the overall RNA-seq data, for example, what is the coverage of the sequencing? Whatare the total reads? Why did you you the reference genome of highbush blueberries for your analysis of tetraploid rabbiteye blueberry?

2. Line 105: When you used “false discovery rate <0.01” as a cut-off, the P-values should be all < 0.001 (if you used Edge R). But it seemed that you use P < 0.05 for some DEGs.

3. Line 105: Move Figure 6 here as Figure 1A to make the results easily understood.

4. Line 133: Add a sentence to conclude the results of these data. The same is for the following paragraphs in this section.

5. Line 143: I do not think that fully chilled leaf buds will develop into flower buds and then into flower. In the other words, from leaf buds to flower buds is not a consecutive stage.

6. Line 147: This table can be removed by spell out the names of these seven genes in the text.

7. Line 176: This is not very consistent with what you described below. What are the criteria for being the candidate genes? I think you talked about the "DEGs of major known genes".

8. Line 184: What do these DEGs indicate?

9. Line 189: What is floral stem? What are the DEGS (of the 34) related to flowering of flower bud formation? You need to include the DEGs in a supplemental table with gene names and expression levels.

10. Lines 200-201: DEGs of these phytohormones showed differential expressions. How do you correlate to their biological function as well as their impact on phenotype(s).

11. Line 204: What do you mean "a sizable change"?

12. Lines 212-213: In your M&M, please describe how you identify and analyze these hormone-related genes? It is hard to believe that this statement "Genes related to cytokinins and auxin were up-regulated while genes related to ethylene were down-regulated". Please double check.

13. Line 225: Confusing... This is a rare term.

14. Line 226: COL9 is similar to the CO gene that responds to photoperiod (an upstream gene of FT).

15. Line 242: This figure is very confusing. Please re-label the X-axis, For example, Bud (S2/S1), Initial (S3/S2), ...... What is the inner control for your qRT-PCR?

Discussion: Please rewrite this part by moving the description of your results to the result part.

M&M: How did you collect the tissues? What are the biological controls?

1. Line 357: How did you collect the tissues? What are the biological controls?

2. Line 372: What was the RIN?

3. Line 376: Paired-end sequencing?

4. Line 380: is Q < 20 too low? Q < 30 is often used.

5. Line 388: If you used FDR < 0.01, P value is < 0.001 (if you used edge R).

6. Lines 402-403: Please include the primers in a supplemental table.

7. Lines 418-423: This is interesting. But I do not see much in your result part.

References: Please replace the unnecessary references with some more related ones. There are quite a few recent publications on blueberry flowering, blueberry genome, blueberry transcriptome analysis that are missing.

Figure 6: A scale bar is needed. For the S4 flowers, were they pollinated? Were the basal parts of the flowers included for your analysis? It is not shown in this image.

Reviewer #2: PONE-D-21-18707

The manuscript entitled “"Transcriptome analysis of genes associated with flowering and fruit set in rabbiteye blueberry (Vaccinium ashei).” aims to unravel the molecular mechanisms of flowering and fruit set from vegetative growth to reproductive transition by evaluating the transcriptome at six developmental stages. Given the lack of RNAseq studies at the flowering stage in blueberry, this study will provide a good genomic resource for the community. However, for this study be useful, reproducible, and publishable, the authors should make the sequencing data available by depositing into public databases such as SRA, and provide more detailed results and methodology. Therefore, I would recommend “major” review and publication dependent on the availability of data and results.

Please, find my suggestions and comments below.

Title: Please, make it clear in the title that the study is for understanding the transition in developmental stages.

Introduction: Authors should describe more about the species and how the study can be useful. Rabitteye (V. ashei) is not the main cultivated species of blueberry.

P3-L67: Cite previous transcriptomic studies focused on fruit development in blueberry here.

Colle et al. 2019 (DOI: 10.1093/gigascience/giz012)

Gupta et al. 2015 (DOI: 10.1186/s13742-015-0046-9)

P5-L96: When the authors say “Our results may be more accurate than those from previous analyses employing Trinity”, what previous analyses are you referring to?

P5-L96: Is the advantage regarding the software (Cufflinks over Trinity), or the methodology (use a reference genome over de novo assembly)?

P5-L103: Authors should provide a supplementary material with the differential expression results of each contrast, showing the statistical description, significance, and annotation for each gene.

P5-L113: Colle et al. (2019) also compared pad and cup stages of berry development. A discussion comparing the results will be interesting.

Figure 1: Did the authors performed an all-against-all analyses to generate the Venn-diagram? It is not described in the M&M, nor results were reported.

Figure2: The authors described it as “PCA of variability in DEGs”. Did you use only the DEG for the PCA construction? Please, describe in the material and methods how the PCA was performed.

P6-Line 132: Looking at the PCA clustering and the low number of genes DE between the leaf buds and flower buds, could it be a case of accidental missampling or misidentification?

P7-L144: Authors said that “Comparing S3 with S2, 8043 DEGs were …”. This number of DEG was not reported before.

Figure 4 and 5. Please add the correspondent gene abbreviation on both plots to facilitate the comparison of results.

Figure 6. For me, Figure 6 should be the first figure of the paper to show the different stages being compared.

P12-L265: The GO term “aromatic compound catabolic process” is not related to aroma compounds. It is related to any substance containing an aromatic carbon ring. Please, rewrite the discussion.

P15-L325: Do you have any other evidence that blueberries are more vulnerable to disease at the cup stage? Probably there are other hundreds of disease-related genes in the genome.

P12-L255: The reference 11 is not a transcriptomic study of flowering.

P17-369: Please add a table with the sample name, number of raw reads, number of mapped reads, number of genes with reads mapped, and accession number on public databases (such as SRA).

P17-L376: Please, add more details about the RNAseq data, if paired end, read length, unstranded.

P18-L378. Cite Colle et al. (2019) as the source of the genome as well.

P18-L384: Please clarify the differential expression analyses pipeline. Did you map the reads against the genome or assembled them? How was the tetraploid phased genome handled regarding multiple mapping reads? Was a multiple testing correction applied?

P18-L390: Did you perform only GO annotation for DEG or a GO enrichment analyses? Was it performed only for unique DEGs or for all DEG? Please, clarify.

P18-L397: What criteria was used to select the 9 transcripts for qRT-PCR?

P19-L401. Please, provide a table or a supplementary material with the primer sequence for each gene and control used.

6. PLOS authors have the option to publish the peer review history of their article (what does this mean?). If published, this will include your full peer review and any attached files.

Reviewer #1: No

Reviewer #2: No

---

## [Author Response · Author response to Decision Letter 0]

7 Aug 2021

Dear Prof. Sussmilch

Re: Manuscript reference PONE-D-21-18707

Please find attached a revised version of our manuscript “Transcriptome analysis and identification of genes associated with floral transition and fruit development in rabbiteye blueberry (Vaccinium ashei)”, which we would like to resubmit for publication as a research paper in PLOS ONE.

Your comments, as well as those of the communicating editor and reviewers, were highly insightful and enabled us to greatly improve the quality of our manuscript. At the end of this file, we provide a point-by-point response to all the comments we received.

We have extensively revised the original manuscript, and the changes can be found in red color font in the text. We hope that the revisions in the manuscript and our accompanying responses will be sufficient to make our manuscript suitable for publication in PLOS ONE.

We look forward to hearing from you at your earliest convenience. 

Yours sincerely,

Xuan Gao, Prof. Dr.

College of Life Sciences, Anhu Normal University

Wuhu, Anhui Province, China

E-mail: gaoxuan@ahnu.edu.cn

---

## [Decision Letter · Decision Letter 1]

16 Aug 2021

PONE-D-21-18707R1

Transcriptome analysis and identification of genes associated with floral transition and fruit development in rabbiteye blueberry (Vaccinium ashei)

PLOS ONE

Dear Dr. gao,

Thank you for submitting your manuscript to PLOS ONE. After careful consideration, we feel that it has merit but does not fully meet PLOS ONE’s publication criteria as it currently stands. Therefore, we invite you to submit a revised version of the manuscript that addresses the points raised during the review process.

Many of the initial reviewer concerns have been addressed in this revised manuscript, and the manuscript has been improved considerably. However, in comments to the Editor, one reviewer expressed valid concerns that the analyses and statistical methods remain insufficiently described including:

filter parameters (both Q<20 and Q<30 are listed in the method and Table S1, with no explanation of which was selected to obtain “the high-quality reads” used in the analyses),statistical significance cutoffs (p<0.05 or p<0.01? both are listed in the methods without explanation),adjusted p-values are lacking from tables in the supporting information,methods for how the PCA and enrichment analyses were performed, andthe identity of reference genes for RT-qPCR (two different genes mentioned in the results and supporting information; how was the gene/s selected and shown to be stably expressed?).

There was also a concern raised about the misidentification of leaf and flower buds – this issue may affect the relevance of comparisons. Perhaps the authors could consider checking the expression of flower or leaf-specific marker genes to determine the extent of this problem and/or consider combining these samples for or removing them from comparisons? The methods do not describe how many leaf and flower buds (or other tissue types) were grouped per replicate.

In addition, there was a concern raised that results were overinterpreted in the discussion.

There are also issues with logFC values in the latter rows of the tables in the supporting information.

Based on these issues, I am recommending a second round of major revisions to give the opportunity for these to be addressed.

We look forward to receiving your revised manuscript.

Kind regards,

Frances Sussmilch

Academic Editor

PLOS ONE

Reviewers' comments:

Reviewer's Responses to Questions

**Comments to the Author**

1. If the authors have adequately addressed your comments raised in a previous round of review and you feel that this manuscript is now acceptable for publication, you may indicate that here to bypass the “Comments to the Author” section, enter your conflict of interest statement in the “Confidential to Editor” section, and submit your "Accept" recommendation.

Reviewer #1: (No Response)

Reviewer #2: (No Response)

2. Is the manuscript technically sound, and do the data support the conclusions?

Reviewer #1: (No Response)

Reviewer #2: Partly

3. Has the statistical analysis been performed appropriately and rigorously? 

Reviewer #1: (No Response)

Reviewer #2: No

4. Have the authors made all data underlying the findings in their manuscript fully available?

Reviewer #1: (No Response)

Reviewer #2: Yes

5. Is the manuscript presented in an intelligible fashion and written in standard English?

Reviewer #1: (No Response)

Reviewer #2: Yes

6. Review Comments to the Author

Reviewer #1: (No Response)

Reviewer #2: (No Response)

7. PLOS authors have the option to publish the peer review history of their article (what does this mean?). If published, this will include your full peer review and any attached files.

Reviewer #1: No

Reviewer #2: No

---

## [Author Response · Author response to Decision Letter 1]

7 Sep 2021

Dear Prof. Sussmilch

Re: Manuscript reference PONE-D-21-18707R1

Please find attached a revised version of our manuscript “Transcriptome analysis and identification of genes associated with floral transition and fruit development in rabbiteye blueberry (Vaccinium ashei)”, which we would like to resubmit for publication as a research paper in PLOS ONE.

Your comments, as well as those of the communicating editor and reviewers, were highly insightful and enabled us to greatly improve the quality of our manuscript. In the following pages, we provide a point-by-point response to all the comments we received.

We have extensively revised the original manuscript, and the changes can be found in red color font in the text. We hope that the revisions in the manuscript and our accompanying responses will be sufficient to make our manuscript suitable for publication in PLOS ONE.

We look forward to hearing from you at your earliest convenience. 

Yours sincerely,

Xuan Gao, 

College of Life Sciences, Anhu Normal University

Wuhu, Anhui Province, China

E-mail: gaoxuan@ahnu.edu.cn

---

## [Decision Letter · Decision Letter 2]

24 Sep 2021

PONE-D-21-18707R2Transcriptome analysis and identification of genes associated with floral transition and fruit development in rabbiteye blueberry (Vaccinium ashei)PLOS ONE

Dear Dr. gao,

Thank you for submitting your manuscript to PLOS ONE. A few minor points remain to be addressed in order for your manuscript to meet PLOS ONE’s publication criteria. Therefore, we invite you to submit a revised version of the manuscript that addresses the points raised during the review process.

As Reviewer 2 was unavailable to review this revised version of your manuscript, a third reviewer was invited - please address Reviewer 3’s comments. If you are not in a position to perform the additional experiments examining anthocyanin levels, consider altering your wording to address Reviewer 3’s 4^th^ comment. Please include the SRA accession number for your data (PRJNA745351 is reported in your cover letter) within the text of your manuscript.

We look forward to receiving your revised manuscript.

Kind regards,

Frances Sussmilch

Academic Editor

PLOS ONE

Journal Requirements:

Reviewers' comments:

Reviewer's Responses to Questions

**Comments to the Author**

1. If the authors have adequately addressed your comments raised in a previous round of review and you feel that this manuscript is now acceptable for publication, you may indicate that here to bypass the “Comments to the Author” section, enter your conflict of interest statement in the “Confidential to Editor” section, and submit your "Accept" recommendation.

Reviewer #1: (No Response)

Reviewer #3: (No Response)

2. Is the manuscript technically sound, and do the data support the conclusions?

Reviewer #1: (No Response)

Reviewer #3: Yes

3. Has the statistical analysis been performed appropriately and rigorously? 

Reviewer #1: (No Response)

Reviewer #3: Yes

4. Have the authors made all data underlying the findings in their manuscript fully available?

Reviewer #1: (No Response)

Reviewer #3: No

5. Is the manuscript presented in an intelligible fashion and written in standard English?

Reviewer #1: (No Response)

Reviewer #3: Yes

6. Review Comments to the Author

Reviewer #1: (No Response)

Reviewer #3: Dear Prof. Sussmilch，

Thank you for giving me a chance to peer review manuscript (No. PONE-D-21-18707_R2) for PLOS ONE. The manuscript entitled “Transcriptome analysis and identification of genes associated with floral transition and fruit development in rabbiteye blueberry (Vaccinium ashei)” described the transcriptomic changes of rabbiteye blueberry flower development from bud to fruit. A great number of DEGs were found to be enriched in phytohormone, transporter protein, photosynthesis, anthocyanins biosynthesis, disease resistance protein and transcription factors.

The study is interesting. And the transition process of rabbiteye blueberry from bud to fruit is important. The justification for this work is to understand the molecular regulatory mechanism of this transition process. However, I still do have some issues with this manuscript, especially in the sections of Abstact, Discussion and Materials and Methods.

1. Page 20-Line 437: Were the samples S2 used for RNA-seq analysis whole flower bud? As far as I know, blueberry flowers are inflorescence. The developmental stages of flowers on the top and bottom might be different, so the replicated samples had not gathered together.

2. Page 20-Line 441: This statement about "……and the basal parts of the flowers were not included" is ambiguous. Were "the basal parts of the flowers" carpopodia or petals?

3. Page 2-Lines 29-30, P11-Line 252, P12-Lines 267-268: "….. these DEGs were mostly enriched in phytohormone, ….". The statament is ambiguous. Please define what pathways of DEGs were enriched, biosynthesis, metabolism, or signal transduction?

4. Since the authors states that anthocyanins started to accumulate from early fruit stage. The level of anthocyanin should be measured in different development stages, at least stages S5 and S6, which is very necessary and important evidence. In fact, our unpublished data showed that the levels of anthocyanin were too low to be dected before pigments were initially accumulated in the highbush blueberry fruit peel.

5. "The PLOS Data policy requires authors to make all data underlying the findings described in their manuscript fully available without restriction, with rare exception (please refer to the Data Availability Statement in the manuscript PDF file). The data should be provided as part of the manuscript or its supporting information, or deposited to a public repository." However, I have not found supporting information from the revised manuscript. Please deposit the sequencing data generated in this study in a suitable public repository such as the NCBI SRA database. Once the sequencing data have been deposited, please provide the information on deposition and how to access these data, including the permanent link or the unique identifier associated to it.

7. PLOS authors have the option to publish the peer review history of their article (what does this mean?). If published, this will include your full peer review and any attached files.

Reviewer #1: No

Reviewer #3: No

---

## [Author Response · Author response to Decision Letter 2]

9 Oct 2021

Dear Dr. Sussmilch:

 On behalf of my co-authors, we thank you very much for giving us an opportunity to revise our manuscript, we appreciate editor and reviewers very much for their positive and constructive comments and suggestions on our manuscript. 

 We have studied reviewer's comments carrefully and have made revision which marked in the paper. We have tried our best to revise our manuscripts according to the comments. Attached please find the respond to reviewers.

 We would like to express our great appreciation to you and reviewers for comments on our paper. Looking forward to hearing from you.

Thank you and best regards.

Yours sincerely,

Xuan Gao

---

## [Editor Report · Decision Letter 3]

13 Oct 2021

Transcriptome analysis and identification of genes associated with floral transition and fruit development in rabbiteye blueberry (Vaccinium ashei)

PONE-D-21-18707R3

Dear Dr. gao,

We’re pleased to inform you that your manuscript has been judged scientifically suitable for publication and will be formally accepted for publication once it meets all outstanding technical requirements.

Kind regards,

Frances Sussmilch

Academic Editor

PLOS ONE
---

## [Editor Report · Acceptance letter]

20 Oct 2021

PONE-D-21-18707R3 

Transcriptome analysis and identification of genes associated with floral transition and fruit development in rabbiteye blueberry (*Vaccinium ashei*) 

Dear Dr. Gao:

I'm pleased to inform you that your manuscript has been deemed suitable for publication in PLOS ONE. Congratulations! Your manuscript is now with our production department. 

Kind regards, 

on behalf of

Dr. Frances Sussmilch 

Academic Editor

PLOS ONE